# MSE-VGG: A Novel Deep Learning Approach Based on EEG for Rapid Ischemic Stroke Detection

**DOI:** 10.3390/s24134234

**Published:** 2024-06-29

**Authors:** Wei Tong, Weiqi Yue, Fangni Chen, Wei Shi, Lei Zhang, Jian Wan

**Affiliations:** 1School of Information and Electronic Engineering, Zhejiang University of Science and Technology, Hangzhou 310023, China; 222108855018@zust.edu.cn (W.T.); sw1031499931@163.com (W.S.); leizhang@zust.edu.cn (L.Z.); wanjian@zust.edu.cn (J.W.); 2School of Computer Science, Hangzhou Dianzi University, Hangzhou 310018, China; 2343050096@hdu.edu.cn

**Keywords:** ischemic stroke, electroencephalography, feature fusion, deep learning

## Abstract

Ischemic stroke is a type of brain dysfunction caused by pathological changes in the blood vessels of the brain which leads to brain tissue ischemia and hypoxia and ultimately results in cell necrosis. Without timely and effective treatment in the early time window, ischemic stroke can lead to long-term disability and even death. Therefore, rapid detection is crucial in patients with ischemic stroke. In this study, we developed a deep learning model based on fusion features extracted from electroencephalography (EEG) signals for the fast detection of ischemic stroke. Specifically, we recruited 20 ischemic stroke patients who underwent EEG examination during the acute phase of stroke and collected EEG signals from 19 adults with no history of stroke as a control group. Afterwards, we constructed correlation-weighted Phase Lag Index (cwPLI), a novel feature, to explore the synchronization information and functional connectivity between EEG channels. Moreover, the spatio-temporal information from functional connectivity and the nonlinear information from complexity were fused by combining the cwPLI matrix and Sample Entropy (SaEn) together to further improve the discriminative ability of the model. Finally, the novel MSE-VGG network was employed as a classifier to distinguish ischemic stroke from non-ischemic stroke data. Five-fold cross-validation experiments demonstrated that the proposed model possesses excellent performance, with accuracy, sensitivity, and specificity reaching 90.17%, 89.86%, and 90.44%, respectively. Experiments on time consumption verified that the proposed method is superior to other state-of-the-art examinations. This study contributes to the advancement of the rapid detection of ischemic stroke, shedding light on the untapped potential of EEG and demonstrating the efficacy of deep learning in ischemic stroke identification.

## 1. Introduction

Ischemic stroke is an acute cerebrovascular disease caused by pathological changes such as vascular stenosis and thrombosis in the cerebral blood vessels [1,2,3]. It is one of the leading causes of death and long-term disability, thus placing a large burden on society [4,5]. Therefore, rapid identification and early treatment of ischemic stroke are crucial [6,7,8]. As shown in Figure 1, traditional methods for detecting ischemic stroke largely depend on imaging examinations such as computed tomography (CT) and magnetic resonance imaging (MRI) [9,10]. Despite their acceptable accuracy, their high cost and rigorous environmental requirements prevent some patients from receiving timely assistance. In light of this, electroencephalography (EEG) has emerged as a promising alternative for detecting ischemic stroke due to its low cost, real-time process capabilities, non-invasive nature, and accessibility [11,12]. As one of the traditional neurophysiological techniques, EEG can capture normal and abnormal brain activity through electrodes placed on the scalp [13] and can be performed in various settings, such as ambulances, emergency rooms, and clinical sites, as shown in Figure 1. This approach can not only reduce the cost of data collection but also enable the early identification of stroke, thereby giving the opportunity to provide timely and effective treatment to patients [14].

In recent years, scholars have attempted to identify stroke features from EEG signals. Assenza et al. [15] demonstrated that the contralateral δ power in EEG signals is a predictive indicator of diagnostic outcomes on the National Institute of Health Stroke Scale (NIHSS). Building on this, Laura C.C. et al. [16] introduced an innovative EEG-based detection method for major vascular occlusive stroke in emergency room settings, achieving the remarkable diagnostic accuracy of 0.83 by utilizing EEG delta-to-alpha ratio features. Aminov et al. [17] demonstrated that data generated by a single prefrontal electrode supported the prognostic value of acute DAR and identified delta-to-theta ratio (DTR) as a potential marker of cognitive outcomes after stroke. Finnigan et al. [18] developed the acute Delta Change Index (aDCI), a metric calculated from two distinct quantitative EEG assessments performed during the acute phase of ischemic stroke. Further research by Finnigan et al. [19] confirmed a significant positive correlation between the NIHSS score and the DAR (delta-to-alpha power ratio), as well as a significant negative correlation between the NIHSS score and the relative α power. Although research on the rapid identification of ischemic stroke using EEG data has achieved some initial results, the accuracy of these methods is currently not high enough to meet the standards for clinical use.

Although research on EEG-based identification methods in the field of stroke diagnosis is in its early stages, EEG has been widely applied in the diagnosis of diseases such as epilepsy, depression, Alzheimer’s disease, etc. [20,21,22]. Common EEG signal characterization methods mainly include functional connectivity and nonlinear dynamic analysis.

Functional connectivity (FC) is a significant method in EEG characterization research. Normal brain function does not rely on the action of a single brain area but on the collaboration among different brain areas. Therefore, studying the functional connections among different brain areas is crucial to understanding the functional mechanisms of the brain. Functional connectivity indicators are used to measure the degree of correlation between two EEG channels, such as Pearson Correlation Coefficient (PCC) [23], Mutual Information (MI) [24], Phase Locking Value (PLV) [25], and Phase Lag Index (PLI) [26]. These functional connectivity indicators have been widely used in various studies. Specifically, Hong et al. [27] used the PLI as a feature to classify depressed patients and healthy controls with an accuracy of more than 92%. Wang et al. [28] built a drug resistance prediction model for epileptic patients by using the PLI connectivity matrix, and the prediction accuracy reached 94%.

In addition, the brain can be approximated to a nonlinear dynamic system; thus, nonlinear dynamic analysis can be specifically tailored to evaluate the complexity of EEG signals and assess the function and state of the brain by calculating various parameters from the time series of EEG signals. Commonly used nonlinear dynamic parameters include Approximate Entropy (ApEn), Sample Entropy (SaEn), Fuzzy Entropy (FuEn), Lyapunov Exponents, and Correlation Dimensions, all of which have been widely utilized in brain neurological disease prediction in various EEG-based research studies. Catherien et al. [29] used the arrangement entropy of EEG signals to detect attention deficit hyperactivity disorder (ADHD) with an accuracy of 99.82%. Song et al. [30] developed the Optimal Sample Entropy (O-SampEn) algorithm and combined it with an extreme learning machine (ELM) to identify the EEG signals of seizures. Zhang et al. [31] proposed a multidimensional Sample Entropy algorithm, which achieved good results in epilepsy prediction.

To rapidly identify ischemic stroke EEG data, it is necessary not only to perform the feature characterization of EEG signals but also to implement automatic classification based on EEG signals [32,33,34]. Current research primarily utilizes deep learning methods to build automatic classification models based on EEG signals, and promising classification results have been achieved [35,36,37]. Specifically, Erguzel et al. [38] optimized the classification of major depressive disorder (MDD) and normal patients by combining the genetic algorithm and the backpropagation neural network and improved the classification accuracy of the classification model. Li et al. [39] combined the convolutional neural network (CNN) and the long short-term memory (LSTM) network to develop a neural network feature fusion algorithm, which improves accuracy in EEG signal classification. C. Zhang et al. [40] proposed the EEG-Inception neural network to achieve the end-to-end accurate classification of MI EEG signals. Wei et al. [41] introduced the Convolutional Block Attention Module (CBAM) into the CNN model to perform Mild Cognitive Impairment (MCI) and Normal Cognition (NC) classification. Xin Ding et al. [42] combined the CNN with the multi-head attention mechanism to achieve accurate predictions on an epilepsy dataset.

The techniques of feature extraction and automatic classification based on EEG signals have been applied in the diagnosis of various diseases, but their use is not yet widespread in the diagnosis of ischemic stroke. In this study, we aimed to rapidly identify ischemic stroke based on EEG data. Our primary contributions are summarized as follows:A new deep learning model, MSE-VGG, was developed to perform the ischemic stroke detection task based on EEG signals. We constructed the model by introducing multiple squeeze-and-excitation (SE) modules into the traditional VGG model, which enhanced its ability to capture ischemic stroke features from EEG signals, allowing it to effectively distinguish between patients with ischemic stroke and non-stroke individuals.The functional connectivity information implied in EEG channels and complexity information are extracted by using the correlation-weighted Phase Lag Index (cwPLI) and Sample Entropy (SaEn), respectively. A fusion feature is proposed by combining these two types of information, which can not only capture the spatio-temporal relationships among multi-channel EEG electrodes but also learn the nonlinear dynamic changes in EEG signals of ischemic stroke patients.Extensive experiments were conducted on the private ZJU4H dataset to validate our model. The experimental results prove the superior performance of our method in the rapid identification of ischemic stroke based on EEG data, indicating that it is suitable for potential real-time application.

The rest of the paper is organized as follows: In Section 2, we describe the dataset we utilized and the methods proposed, including data preprocessing, feature extraction, and the construction of the classification model. In Section 3, we prove the superiority of the proposed model based on experimental analysis. Finally, in Section 4, we summarize the research results and suggest potential avenues for further exploration.

## 2. Materials and Methods

In this section, data acquisition and the proposed method are illustrated in detail. First, the source and collection of EEG data are described. Then, the rapid ischemic stroke detection method, consisting of preprocessing, feature extraction, and classification, is illustrated.

### 2.1. Data Acquisition

This research study was approved by the Ethics Committee of The Fourth Affiliated Hospital of Zhejiang University School of Medicine. All test results appearing in the original medical records (including personal data, laboratory documents, etc.) will be kept completely confidential to the extent permitted by law. The ZJU4H EEG dataset utilized in this study was derived from The Fourth Affiliated Hospital of Zhejiang University School of Medicine. It includes high-quality EEG data from 20 ischemic stroke patients (11 males and 9 females, aged from 47 to 87 years old) and 19 non-stroke controls (12 males and 7 females, aged from 45 to 76 years old). All EEG signal was collected by XLTEK-EEG32U with 32 channels. The EEG data in this dataset retain 19-channel electroencephalogram signals consistent with the international 10–20 electrode placement standard. The sampling rate was 256 Hz, and approximately 5 min of EEG data was collected per patient. Detailed information on the ZJU4H dataset is summarized in Table 1. MRI diagnosis was the basis for labeling the subjects.

### 2.2. Methods

An EEG-based ischemic stroke detection framework which consists of three main parts, as illustrated in Figure 2, was developed in this study. First, preprocessing and segmentation are performed on raw EEG data as introduced in Section 2.2.1. Second, feature extraction is implemented by calculating the cwPLI and SaEn on the segmented EEG signals. Finally, feature maps are generated by fusing the two information types and converted into RGB figures. The process of feature extraction and feature fusion are given in detail in Section 2.2.2. Finally, the novel MSE-VGG model is constructed for classification as described in Section 2.2.3.

#### 2.2.1. Preprocessing

The main objective of EEG preprocessing is to reduce or remove mixed noise and interference in EEG signals. The preprocessing methods employed in this study include filtering, interpolation of bad channels, and artifact removal.

The EEGLAB toolbox in MATLAB was used to preprocess the EEG signals. Existing research [15,16,17,18,19] has indicated that the mid-to-low frequency bands within electroencephalography (EEG) signals exhibit a more significant correlation with stroke. Therefore, we selected a 1–35 Hz bandpass filter to filter the EEG signals, which can remove 50 Hz power frequency interference while preserving effective information (δ band: 1–4 Hz; θ band: 4–8 Hz; α band: 8–13 Hz; β band: 13–30 Hz). After filtering, the EEG data are checked manually, and spline interpolation is performed on the channels with poor data. During the acquisition of EEG signals, eye movements, blinking, and muscle activity in the head and neck of the patient can interfere with EEG [27]. Therefore, it is necessary to manually remove EMG (electromyography) and EOG (electrooculogram) artifacts to increase the quality of EEG signals. Finally, the latter are segmented by using a 1-s sliding window with no overlap.

#### 2.2.2. Feature Extraction

Recent research [13,43] has provided preliminary results that prove that EEG could be a feasible approach for stroke detection. During ischemic stroke, localized ischemic necrosis of brain tissue causes significant changes in the EEG signals in certain areas of the brain. Since the brain is a complex connecting organ, these changes continue to spread, leading to high correlations between adjacent regions. In this study, to fully utilize the rich spatio-temporal domain information of EEG signals, the Phase Lag Index (PLI) values between different channels, which can efficiently characterize the underlying spatial resolution of EEG signals, were calculated. We used the Hilbert transform to find the instantaneous phase and construct the brain network correlation matrix based on phase synchronization. The PLI is an important parameter for quantifying the degree of phase synchronization between two signals from different channels and is calculated as follows:(1)$$PLI=\left|\langle \mathrm{sign}\left(\Delta \emptyset_{\mathrm{rel}\;}\left(t\right)\right)\rangle \right|=\left|\frac{1}{K}\sum_{k=1}^{K}\mathrm{sign}\left(\Delta \emptyset_{\mathrm{rel}\;}\left(t_{k}\right)\right)\right|$$
where *K* represents the sampling points of each EEG segmentation operation; $\Delta \emptyset_{\mathrm{rel}\;}\left(t_{k}\right)$ represents the phase difference between two signals at time $t_{k}$; and sign is a mathematical operation with 1 when the argument is positive, −1 when the argument is negative, and 0 when the argument is zero. The value range of the PLI is [0,1], and the larger the value, the stronger the degree of phase synchronization between two signals.

Then, a PLI functional correlation matrix of EEG signals is constructed as follows:(2)$$\mathrm{PLI}=\left[\begin{array}{cccc} PLI_{11} & PLI_{12} & \ldots & PLI_{1N}\\ PLI_{21} & PLI_{22} & \ldots & PLI_{2N}\\ \vdots & \vdots & \ddots & \vdots \\ PLI_{N1} & PLI_{N2} & \ldots & PLI_{NN} \end{array}\right]$$

In the matrix, element $PLI_{ij}$ represents the synchronization degree between channel *i* and channel *j*, and N represents the number of EEG channels. Due to the symmetrical characteristic, i.e., $PLI_{ij}=PLI_{ji}$, only the upper-triangle elements in the matrix need to be calculated.

In order to further increase the difference between the two classes (ischemic stroke and non-stroke control) of EEG data and consequently enhance the identification ability of the model, we introduce a novel correlation weight which reveals the correlation between PLI elements and the data class. The correlation weight can be denoted as
(3)$$w_{ij}={\left|\frac{n_{l}-n_{s}}{P\times M}\right|}_{ij}$$
where $n_{l}$ is the number of ischemic stroke patients with $PLI_{ij}$ greater than that of non-stroke controls, $n_{s}$ is the number of ischemic stroke patients with $PLI_{ij}$ smaller than that of non-stroke controls, *P* is the number of ischemic stroke patients, and *M* is the number of non-stroke controls. Since the relationship among subjects belonging to the same group is not taken into account, the total number of connectivity comparisons to be calculated is P×M.

The correlation weight $w_{ij}$ indicates the correlation between the feature $PLI_{ij}$ and the classification (stroke or non-stroke). It can also expand the connectivity difference between two classes of EEG data. We construct the correlation-weighted PLI matrix, i.e., cwPLI. The matrix of the correlation-weighted Phase Lag Index is denoted by
(4)$$\mathrm{cwPLI}=\left[\begin{array}{cccc} PLI_{11}\times w_{11} & PLI_{12}\times w_{12} & \ldots & PLI_{1N}\times w_{1N}\\ PLI_{21}\times w_{21} & PLI_{22}\times w_{22} & \ldots & PLI_{2N}\times w_{2N}\\ \vdots & \vdots & \ddots & \vdots \\ PLI_{N1}\times w_{N1} & PLI_{N2}\times w_{N2} & \ldots & PLI_{NN}\times w_{NN} \end{array}\right]$$

Sample Entropy (SaEn) [44] characterizes the complexity of a time series by quantifying the probability of generating new patterns, reflecting the randomness and regularity of EEG signals. Therefore, compared with other time-domain features (such as mean, variance, etc.), Sample Entropy provides a more accurate description of EEG signals. The calculation process of SaEn for the signal $X^{K}=\left\{x\left(1\right),x\left(2\right),\ldots ,x\left(K\right)\right\}$ is divided into 5 steps:

(1) Define the integers *m* and *r*, where *m* represents the length of the comparison vector and *r* represents the measure of similarity.

(2) Reconstruct the original sequence for the m-dimensional vector as follows:(5)$$X^{m}=\left\{x\left(1\right),x\left(2\right),\ldots ,x\left(K-m+1\right)\right\},$$
where
(6)$$\begin{array}{c} X^{m}\left(i\right)=\left\{x\left(i\right),x\left(i+1\right),\ldots ,x\left(i+m-1\right)\right\},i=1,2,\ldots ,K-m+1 \end{array}$$

(3) Calculate the Euclidean distance $d[X^{m}\left(i\right),X^{m}\left(j\right)]$ between any vector $X^{m}\left(i\right)$ and the rest of the vectors as follows:(7)$$d\left[X^{m}\left(i\right),X^{m}\left(j\right)\right]=\max_{k\in (1,m-1)}\left|x\left(i+k\right)-x\left(j+k\right)\right|$$

(4) Determine the number $K^{m}\left(i\right)$ that satisfies $d[X^{m}\left(i\right),X^{m}\left(j\right)]<r$ for each vector $X^{m}\left(i\right)$, and calculate the ratio $B_{i}^{m}\left(r\right)$ between the number $K^{m}\left(i\right)$ and the total number of the rest of the vectors K−m−1 as follows:(8)$$B_{i}^{m}\left(r\right)=\frac{N^{m}\left(i\right)}{K-m-1}$$

(5) SaEn [45] can be derived with the following equation:(9)$$SaEn\left(m,r,K\right)=-\ln\left[\frac{B^{m+1}\left(r\right)}{B^{m}\left(r\right)}\right]$$

After calculating the cwPLI matrix and SaEn, the upper triangle of the cwPLI matrix is fused with the SaEn array to obtain a two-dimensional fusion feature, which is then converted into a RGB map as the final feature fed to the classification model. Algorithm 1 summarizes the detailed steps of feature extraction.
**Algorithm 1** Algorithm for feature extraction**Input:** EEG database $N=N_{n}^{s}$, where *n* is the number of EEG channels, *s* means the number of subjects.**Output:** RGB map of cwPLI+SaEn   **for**
p=1 to s **do**      preprocess the EEG signal      dataset $D=D^{d}\leftarrow$ segmented each $N_{n}^{p}$ into epochs of 1 s      ${\mathrm{PLI}}^{p}\leftarrow$ calculate the PLI matrix of each $N_{n}^{p}$      **for** i=1 to n **do**        **for** j=1 to n **do**           $w_{ij}\leftarrow$ calculate the correlation weight of each $PLI_{ij}$ in ${\mathrm{PLI}}^{p}$ matrix        **end for**      **end for**      $w\leftarrow w_{ij}$   **end for**   **for**
m=1 to d **do**      ${\mathrm{PLI}}^{m}\leftarrow$ calculate the PLI of each $D^{m}$      ${\mathrm{cwPLI}}^{m}\leftarrow {\mathrm{PLI}}^{m}\times w$      $SaEn^{m}\leftarrow$ calculate the SaEn of each $D^{m}$      (cwPLI+SaEn)${\;}^{m}\leftarrow {\mathrm{PLI}}^{m}+SaEn^{m}$   **end for**   Convert cwPLI+SaEn to RGB map   **return** Outputs

#### 2.2.3. The MSE-VGG Model

The MSE-VGG model established in this study is an improved version of the VGG neural network [46] incorporating multiple squeeze-and-excitation (SE) modules, specifically designed for the classification of EEG signals. By integrating SE attention modules, the network is able to focus more on important channels and spatial features, thereby enhancing accuracy and robustness in the classification task. As shown in Figure 3, the structure of the MSE-VGG model includes 13 convolutional layers, 3 fully connected layers, and 5 pooling layers, the same as VGG. However, we added one squeeze-and-excitation (SE) block after each pooling layer, which enables the network to focus on key channels more effectively. The model leverages the classic 3×3 convolutional kernel characteristic of the VGG architecture for its convolutional layers, ensuring robust feature extraction. These kernels, coupled with the model’s extended depth, are meticulously designed to enhance the network’s ability to capture complex nonlinear patterns. Furthermore, the incorporation of the ReLU (Rectified Linear Unit) activation function within the convolutional layers accelerates the model’s ability to interpret nonlinear image features, thereby boosting its overall efficiency and performance. The pooling layers strategically utilize max pooling to efficiently downsample the feature maps, preserving the most salient information while reducing the computational load. This well-orchestrated combination of architectural choices propels the model towards superior feature representation and more effective learning.

Therefore, the highlight of our model is the introduction of multiple squeeze-and-excitation (SE) modules based on the VGG model. The lightweight attention structure based on squeeze-and-excitation (SE) blocks [47] is an efficient channel attention mechanism in deep learning networks. The structure of an SE block, as depicted in Figure 4, comprises two primary parts: compression and activation. Initially, based on global average pooling, the input feature maps within each channel undergo compression to derive a global descriptor along the channel dimension. This process contributes to lowering the computational load while preserving comprehensive channel-specific information. Next, the condensed global descriptors are input into a finite Multi-Layer Perceptron (MLP), typically incorporating a hidden layer and an output layer. To reduce dimensions and introduce nonlinear transformations, the hidden layer utilizes ReLU or other activation functions to activate its output. On the other hand, the output layer normally uses the Sigmoid function as the activation function of the output layer to obtain a weight vector between 0 and 1, which is used to adjust the original feature map. Finally, the resulting weights are multiplied by the original feature map to highlight channel or spatial features with higher weights. The detailed calculation steps are as follows: (1) Feature graph X becomes feature graph U after $F_{tr}$. $F_{tr}$ can be thought of as a standard convolutional operator. The formula is defined as follows:(10)$$u_{c}=v_{c}*X=\sum_{s=1}^{C^{\prime }}v_{c}^{s}*x^{s}$$
where X∈
$R^{\left(H^{\prime }\times W^{\prime }\times C^{\prime }\right)}$ is the input feature graph, U∈
$R^{\left(H\times W\times C\right)}$ is the output feature graph, *v* represents a learned set of filter kernels, $v_{c}$ refers to the parameter of the *c* filter, $v_{c}^{s}$ represents a 2D space kernel, and ∗ represents the convolution operation.

(2) By performing the global averaging pooling of channels, the feature graph W×H×C containing global information is directly compressed into a 1×1×C feature vector Z, and the channel features of C feature maps are compressed into a numerical value:(11)$$z_{c}=F_{sq}\left(u_{c}\right)=\frac{1}{H\times W}\sum_{i=1}^{H}\sum_{j=1}^{W}u_{c}\left(i,j\right)$$
where $z_{c}$ denotes the c-th element of *Z*.

(3) Next, the channel dependencies are fully captured by using the excitation operation
(12)$$s=F_{ex}\left(z,W\right)=\delta \left(g\left(z,W\right)\right)=\delta \left(W_{2}\delta \left(W_{1z}\right)\right)$$
where δ stands for the ReLU function, $W_{1}\in R^{\frac{C}{r}\times C}$, and $W_{2}\in R^{C\times \frac{C}{r}}$

(4) The final output is obtained by converting the output U by using activation rescaling:(13)$${\tilde{x}}_{c}=F_{\mathrm{scale}\;}\left(u_{c},s_{c}\right)=s_{c}\cdot u_{c}$$
where $\tilde{X}=\left[{\tilde{x}}_{1},{\tilde{x}}_{2},\ldots ,{\tilde{x}}_{C}\right]$ and $F_{scale}=\left(u_{c},s_{c}\right)$ represents the channel-wise multiplication between the feature graph $u_{c}\in R^{H\times W}$ and the scalar $s_{c}$.

## 3. Experiments and Results

In this section, the superiority of our methods is proven. Initially, the environmental requirements and evaluation metrics are illustrated. Then, the series of comprehensive and comparative experiments we performed are reported. First, the significance of our cwPLI feature in stroke identification was verified. Second, by conducting stroke recognition experiments targeting different features, the effectiveness of the fusion feature including cwPLI and SaEn was confirmed. Next, the superior classification performance of the proposed model was verified by comparing it with other classification models. Further ablation experiments validated the effectiveness of the upgraded model. Finally, an extensive analysis was conducted to illustrate the advantages of the model in terms of portability, low cost, ease of operation, and reduction in time cost.

### 3.1. Experimental Requirements and Metrics

All the experiments were conducted with an RTX TITAN GPU equipped with 32 GB of random access memory (RAM), based on Python 3.8, TensorFlow-GPU version 2.6.0, mne version 1.5.1, and Keras version 2.6.0. The model was trained by using a stochastic gradient descent (SGD) optimizer with a batch size of 512, an initial learning rate of 0.0001, and cross-entropy as the loss function of the reverse propagation gradient. The training session ended at epoch 100. In addition, we used the early stop mechanism to prevent overfitting, and a random number seed was set to ensure the complete reproducibility of the experimental results. During the experimental process, the EEG data from patients with ischemic stroke and the non-stroke control group were labeled as 1 and 0, respectively, for the purpose of classification.

When using a model for prediction, there are four possible relationships between the predicted outcomes and the actual conditions of the subjects:True positive (TP): The model correctly predicts a positive case, i.e., the prediction matches the actual positive conditions.False negative (FN): The model incorrectly predicts a negative case, i.e., the prediction does not match the actual positive conditions.False positive (FP): The model erroneously predicts a positive case, i.e., the prediction does not match the actual negative conditions.True negative (TN): The model correctly predicts a negative case, i.e., the prediction matches the actual negative conditions.

The distinction between ischemic stroke patients and non-stroke controls can be seen as a dichotomous problem. Accuracy (Acc), sensitivity (Sen), and specificity (Spe) are commonly used as dichotomous indicators in the medical field.

Acc is one of the common evaluation indexes in binary classification and refers to the proportion of samples predicted to be accurate out of all samples. The calculation of Acc is as follows:(14)$$\mathrm{Acc}=\frac{\mathrm{TP}+\mathrm{TN}}{\mathrm{TP}+\mathrm{TN}+\mathrm{FP}+\mathrm{FN}}$$

Sen is the percentage of the positive samples that are truly positive. If the sensitivity of the proposed model is too low, an ischemic stroke subject may be diagnosed as a healthy individual, so stroke detection algorithms need to have high sensitivity. The calculation is as follows:(15)$$\mathrm{Sen}=\frac{\mathrm{TP}}{\mathrm{TP}+\mathrm{FN}}$$

Spe is the percentage of true-negative samples within a negative sample, and poor specificity increases the false-positive rate. A false positive may be diagnosed as ischemic stroke in a healthy individual, resulting in a waste of medical resources. Therefore, stroke detection algorithms need to have high specificity. Spe can be obtained as follows:(16)$$\mathrm{Spe}=\frac{\mathrm{TN}}{\mathrm{TN}+\mathrm{FP}}$$

### 3.2. Verification of Proposed cwPLI Feature

In this subsection, the effectiveness of the proposed feature, cwPLI, is evaluated. The EEG signals used were collected based on the international 10–20 electrode system, which consists of 19 channels. The cwPLI results of ischemic stroke patients and non-stroke controls are shown in Figure 5. Both the horizontal and vertical coordinates represent EEG channels, and the different colors represent the magnitudes of the elements of cwPLI, i.e., the cwPLI between the channels. It can be seen that in general, the cwPLI of the non-stroke controls is greater than that of the stroke patients. Figure 6 shows the box diagram of the average PLI and cwPLI. Since the correlation weights range from 0 to 1, the cwPLI has a smaller value than the PLI. Moreover, both the PLI and cwPLI show that the functional connection of non-stroke controls is stronger than that of ischemic stroke patients. The decrease in the PLI is indicative of the compromised synchronization within EEG signals, a consequence of the brain’s functional disruption caused by the rupture of a cerebral blood vessel. The diminished PLI values underscore the impact of ischemic stroke on the brain’s capacity for synchronized neural activity, highlighting the diagnostic potential of the PLI as a sensitive biomarker for stroke. In addition, compared with the PLI, the cwPLI further widened the gap between the values of the non-stroke control and ischemic stroke groups. In other words, the constructed feature cwPLI has superior ability to identify ischemic stroke.

To further explore the differences in functional connectivity between the ischemic stroke patients and non-stroke controls, we visualized the functional connectivity based on the cwPLI. Figure 7 shows the distribution of the functional connections for the ischemic stroke patients and non-stroke controls, using 0.10 as the threshold. The connections are significantly sparser in the ischemic stroke group than in the non-stroke control group. This result once again verifies the superiority of the cwPLI as a feature for stroke identification.

### 3.3. Results

#### 3.3.1. Comparative Analysis of Different Features

To validate the superior performance of the fusion feature which combines the cwPLI and SaEn together, four baseline features, i.e., PCC (Pearson Correlation Coefficient), MI (Mutual Information), PLV (Phase Locking Value), and PLI (Phase Lag Index), were considered for comparison. These four baselines are typical features used to characterize functional connectivity:PCC [23] determines whether there is a positive or negative correlation between two signals and the strength of that correlation.MI [24] is used to measure the amount of information about one signal contained in another and is capable of detecting both linear and nonlinear correlations between two signals.PLV [25] is a phase-based functional connectivity method used to measure the degree of phase synchronization between two channel signals, with higher values indicating stronger synchronization.PLI [26] measures the phase synchronization between two channel signals and is not sensitive to volume conduction effects but may be sensitive to noise.

The experimental results (as shown in Table 2) indicate that the fusion feature cwPLI+SaEn performed best in the MSE-VGG classifier, with accuracy of 90.17%, sensitivity of 89.86%, and specificity of 90.41%. Compared with the PLI feature, cwPLI+SaEn showed increases of 6.07%, 9.84%, and 1.8% in accuracy, sensitivity, and specificity, respectively. Among the four baselines, the PLI was the most sensitive to the EEG signals of ischemic stroke, with the best accuracy, sensitivity, and specificity, further proving the strong identification capability of cwPLI+SaEn. Moreover, the classification results of cwPLI+SaEn show increases of 1.37%, 1.37%, and 1.33% in accuracy, sensitivity, and specificity, respectively, compared with the cwPLI alone. This indicates that the fusion feature which combines the functional connectivity and complexity of EEG signals is the most competent in distinguishing patients with ischemic stroke from non-stroke controls.

In order to show the superiority of the fusion feature cwPLI+SaEn more clearly, the ROC curves of the MSE-VGG classifier with different EEG features are given in Figure 8. The figure clearly demonstrates that the AUC values of the cwPLI feature and the combined cwPLI+SaEn feature are superior to the other four types of features. This indicates that the proposed cwPLI feature extraction algorithm is adept at capturing the functional connectivity differences between ischemic stroke patients and healthy controls, enabling the accurate identification of ischemic stroke patients. Additionally, the integration of Sample Entropy (SaEn) also enhances the identification ability of the cwPLI, enabling the fusion feature (cwPLI+SaEn) to achieve the highest AUC.

#### 3.3.2. Comparative Analysis of Different Classifiers

To evaluate the classification performance of the MSE-VGG model, we compared the results with the following existing popular classifiers:Logistic Regression (LR) [48]: A classical linear discrimination model widely used for various classification tasks.AlexNet [49]: An iconic deep convolutional neural network known for its groundbreaking results in the 2012 ImageNet competition, marking a significant milestone in the field of deep learning for image recognition.RCF (Richer Convolutional Features) [50]: An improved method based on the VGG16 framework, aiming to enhance edge detection accuracy by capturing multi-scale and multi-level information from images.LSTM (Long Short-Term Memory) [51]: A special type of recurrent neural network structure that excels at handling and predicting long-term dependencies in time-series data.VGG [46]: A deep convolutional network that improves image recognition accuracy by using multiple small convolutional kernels (3×3) and increasing the number of network layers, widely applied in the field of computer vision.

The experimental results (see Table 3 for details) show that the MSE-VGG model demonstrates significant advantages over the other five classifiers when classifying EEG signals from ischemic stroke patients and non-stroke controls. For instance, in terms of accuracy, the MSE-VGG model outperforms the LR model by 11.86%, AlexNet by 4.53%, LSTM by 4.19%, VGG16 by 2.82%, and RCF by 2.6%. Additionally, while the RCF model shows poor specificity, this indicates that despite being an optimization of the VGG model, it is not suitable for the task of ischemic stroke recognition. In contrast, the MSE-VGG model is more sensitive to the functional connectivity and complexity of EEG signals, which is effective in recognizing ischemic stroke.

#### 3.3.3. Ablation Experiments

To evaluate the contribution of each component of the proposed ischemic stroke detection framework, we conducted ablation experiments and evaluated the stroke detection results on the ZJU4H dataset. As shown in Table 4, for the feature extraction phase, we compared the PLI feature, the proposed cwPLI feature, and the proposed cwPLI+SaEn combined feature. For the classification phase, we conducted comparative experiments between the standard VGG model and the improved MSE-VGG model, while keeping the classification model unchanged, the classification accuracy based on the cwPLI+SaEn combined feature increased by 2.82% compared with the standalone PLI feature and by 2.38% compared with the standalone cwPLI feature. This emphasizes how the combination of the features cwPLI and SaEn can markedly capture functional connectivity, complexity, and dynamic changes in the EEG signals of ischemic stroke patients and non-stroke controls.

To thoroughly explore the improvements obtained with the proposed MSE-VGG architecture, we performed an additional test to assess the contribution of the multiple SE modules and their impact on stroke detection performance in comparison with the traditional VGG network. With the same feature extraction method, we can see steady performance gains in accuracy, specificity, and sensitivity results as we use efficacious multiple SE modules in the VGG network. For instance, compared with the VGG model, the accuracy of the MSE-VGG model increased by 2.38% based on the cwPLI feature extraction method and by 3.18% based on the cwPLI+SaEn feature extraction method. In other words, the proposed MSE-VGG model can effectively exploit the interchannel dependence and context information of the cwPLI+SaEn feature and efficiently improve stroke detection performance.

A confusion matrix was constructed to visually demonstrate the advantages of the cwPLI+SaEn feature in characterizing EEG signals and the outstanding classification performance of the MSE-VGG model. As shown in Figure 9, label 1 in the confusion matrix represents the EEG of ischemic stroke patients, while label 0 represents the EEG of the healthy control group. In the confusion matrix, the cells with a darker color carry more samples, and the cells on the diagonal indicate the number of samples correctly classified by the model. The results indicate that with the MSE-VGG classifier, the performance of the cwPLI+SaEn feature surpasses that of the cwPLI alone. Moreover, with the feature extraction method set to cwPLI+SaEn, the MSE-VGG model shows stronger recognition ability compared with the traditional VGG model. These outcomes further confirm the potential of the cwPLI+SaEn feature for enhancing accuracy in recognizing ischemic stroke and the effectiveness of the MSE-VGG model in the task of EEG signal classification.

#### 3.3.4. Feasibility Analysis

To evaluate the feasibility of the practical application of the proposed method in ischemic stroke detection, it was compared with traditional stroke detection examinations in terms of portability, cost, operability, and detection time. Traditional examinations included the following:CT (computed tomography): A technology that uses X-rays to penetrate the human body and form images by receiving signals from detectors, providing detailed cross-sectional images to help doctors observe internal structures and abnormalities.MRI (magnetic resonance imaging): A technique that uses a strong magnetic field and radio waves to obtain detailed images of the internal body, suitable for imaging soft tissues, such as brain tissue.fMRI (functional magnetic resonance imaging): A neuroimaging technique that infers brain activity by measuring changes in blood flow during brain function, focusing on brain functional activity rather than anatomical structure.

Fifteen ischemic stroke patients and fifteen healthy controls were randomly selected from the ZJU4H dataset to test our detection method. We recorded the time spent on each subject and calculated the average time as the final result. The results in Table 5 indicate that the proposed method can automatically detect ischemic stroke with only 10 min of EEG signal collection time and 4.62 s of model running time. As we mention in Section 1, EEG devices are portable, compact, and adaptable for use across various scenarios, including ambulances, emergency rooms, and clinical facilities. This portability enables rapid and convenient deployment, a significant advantage over traditional detection methods that are confined to examination rooms and comparatively long setting time, where the time expenditure is further compounded by the transfer to the hospital and the scheduling of medical appointments. Moreover, the proposed EEG-based approach possesses additional benefits, such as portability, ease of use, and cost effectiveness, which enhance its appeal for practical implementation. Its user-friendly nature and affordability make it a more accessible and desirable option for real-world applications, offering a time-efficient and economical alternative to conventional diagnostic procedures.

## 4. Discussion

As we illustrate in Section 1, the existing approaches for stroke detection based on EEG signals are far from having high detection accuracy and struggle to meet clinical requirements. To address this problem, we developed the MSE-VGG model integrated with the cwPLI feature extraction algorithm for the rapid and accurate detection of ischemic stroke based on EEG. Specifically, the cwPLI is a special, weighted version of the PLI which uses the correlation between PLI matrix elements and categories as coefficients. Additionally, the cwPLI can significantly represent the functional connectivity differences between ischemic stroke patients and non-stroke controls, thereby extracting more prominent features. The MSE-VGG model enhances the recognition performance of the VGG model by incorporating squeeze-and-excitation (SE) modules to emphasize the expression of important channels.

The Phase Lag Index (PLI) is one of the commonly used indicators of EEG signal functional connectivity, reflecting the functional network information of the brain. The proposed cwPLI feature, which applies a correlation coefficient to weigh the traditional PLI, offers a more pronounced depiction of the functional connectivity disparities between individuals with ischemic stroke and non-stroke controls (as shown in Figure 6 and Figure 7). Compared with the PLI feature extraction method, models using the cwPLI and cwPLI+SaEn as features have superior ischemic stroke identification capabilities (as shown in Table 2). Moreover, the visual representation of the cwPLI matrix, as depicted in Figure 5, reveals distinct patterns between ischemic stroke patients and non-stroke controls. For the non-stroke group, the cwPLI values are notably higher, manifesting as a more prevalent yellow hue across the distribution map, indicative of stronger functional connectivity. In contrast, the ischemic stroke group exhibits lower cwPLI values, which are reflected in a more widespread blue coloration on the map, signifying reduced connectivity. Therefore, we decided to convert the cwPLI images into a three-channel RGB format for subsequent recognition.

The MSE-VGG model was developed for stroke recognition. Compared with the VGG model, it incorporates multiple SE modules, which learn the weight coefficients of each channel, making the model more discerning of the features of each channel, thereby exhibiting stronger recognition capabilities than the VGG model (as shown in Table 3).

In this study, we developed a novel method for detecting ischemic stroke based on EEG signals. Compared with traditional methods such as CT and MRI, the acquisition of EEG signals has the advantages of convenience, low cost, simple operation, and low time consumption (as shown in Table 5). This convenience allows stroke detection to be performed beyond the hospital environment in outdoor and home settings. Our experimental results show that the EEG-based detection method performs well in diagnosing ischemic stroke, with an accuracy of 90.17% and an AUC value of 0.9013. These results suggest the potential of this method for clinical application and for providing more thorough services to patients.

## 5. Conclusions

In this study, we developed a rapid deep learning detection method for ischemic stroke based on EEG signals. This method takes full advantage of the portability of EEG acquisition equipment and its sensitivity to changes in the brain state, which are at the basis of the contribution of EEG classification to the field of ischemic stroke detection. Furthermore, the proposed MSE-VGG architecture provides the benefits of analyzing fusion features and performing the automatic detection of ischemic stroke. This rapid detection method achieved accuracy, sensitivity, and specificity of 90.17%, 89.86%, and 90.41%, respectively, on the ZJU4H dataset. Compared with traditional stroke detection methods, the proposed model has the advantages of simple operation, low cost, high portability, and short detection time.

The proposed approach is dedicated to establishing an intelligent clinical treatment ecosystem for stroke, offering patients higher-quality diagnostic and therapeutic options. However, the proposed cwPLI feature extraction method is highly dependent on the location of the EEG channels, and only EEG signals from 10 to 20 EEG systems were verified in this study. At the same time, it should be noted that there is still room for accuracy improvements to meet the clinical application standards. Therefore, as part of our future research efforts, we plan to collect more diverse EEG data and construct a larger-scale dataset for verifying our approach. Additionally, we intend to optimize our model to improve its recognition performance.

## Figures and Tables

**Figure 1 sensors-24-04234-f001:**
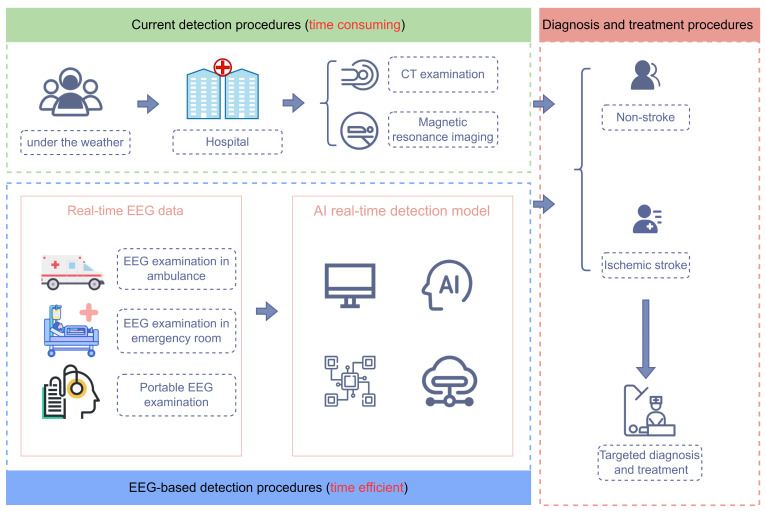
Comparison between traditional and ideal methods for rapid identification of ischemic stroke.

**Figure 2 sensors-24-04234-f002:**
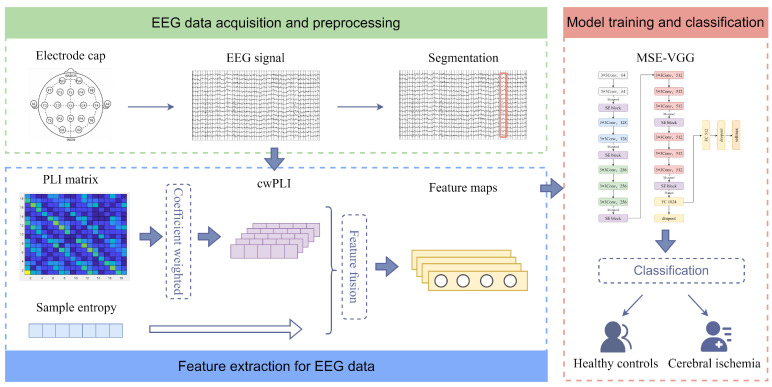
Framework of proposed method for ischemic stroke detection based on EEG signals.

**Figure 3 sensors-24-04234-f003:**
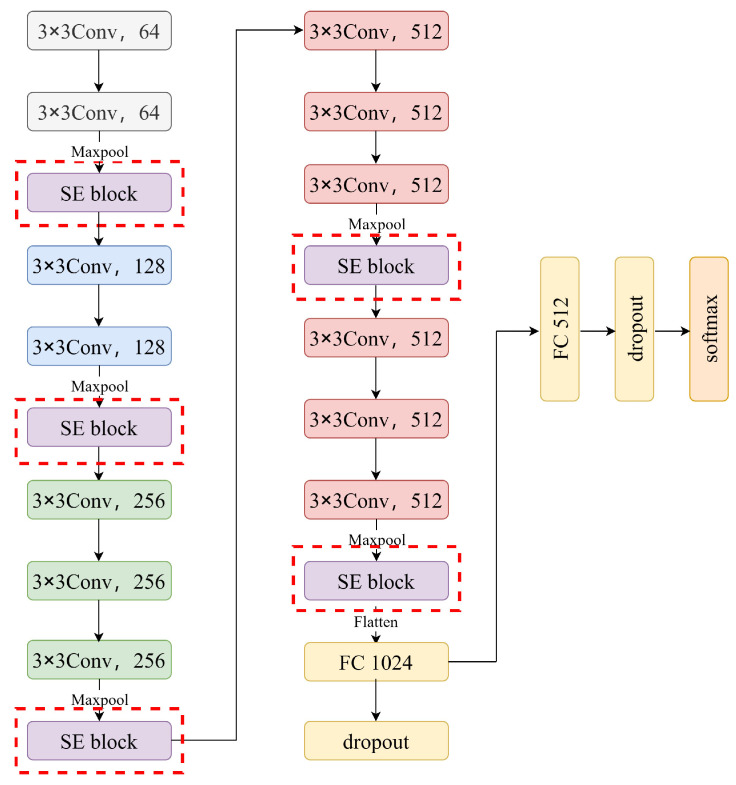
The structure of the proposed MSE-VGG model. The red dashed boxes represent the improvements of the MSE-VGG model to the VGG model.

**Figure 4 sensors-24-04234-f004:**
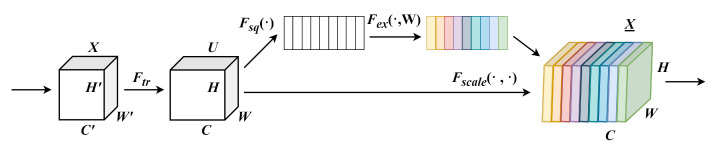
The structure of an SE block.

**Figure 5 sensors-24-04234-f005:**
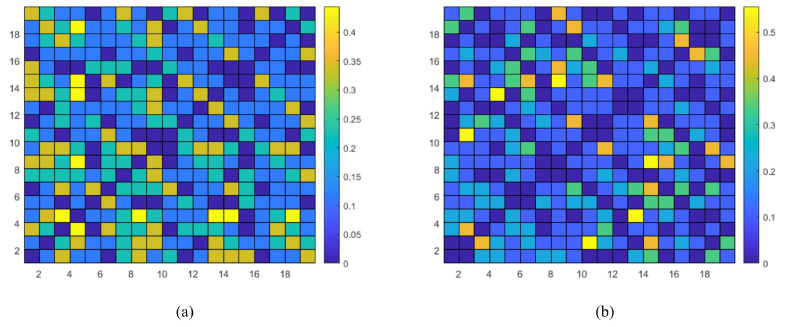
The cwPLI of ischemic stroke patients and non-stroke controls. (**a**) The cwPLI of the non-stroke controls. (**b**) The cwPLI of the ischemic stroke patients.

**Figure 6 sensors-24-04234-f006:**
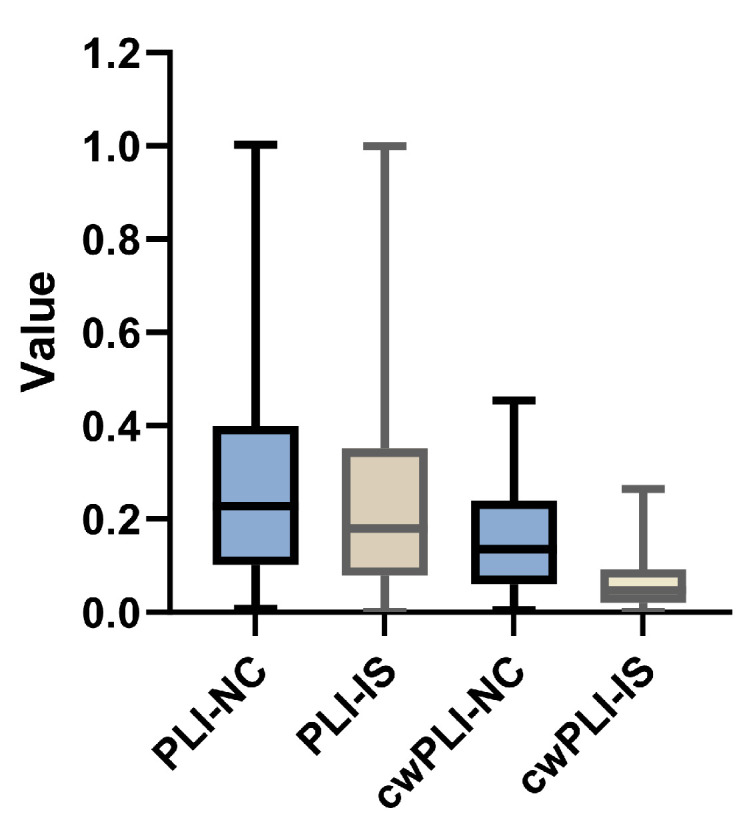
The PLI and cwPLI distributions of ischemic stroke patients and non-stroke controls. PLI-NC represents the PLI value of the non-stroke controls, PLI-IS represents the PLI value of the ischemic stroke patients, cwPLI-NC represents the cwPLI value of the non-stroke controls, and cwPLI-IS represents the cwPLI value of the ischemic stroke patients.

**Figure 7 sensors-24-04234-f007:**
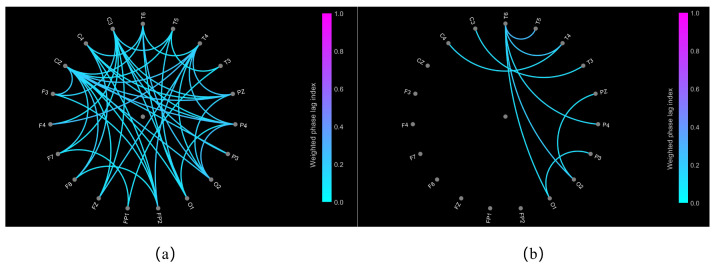
The functional connectivity resolution distributions based on the cwPLI between ischemic stroke patients and non-stroke controls. (**a**) The functional connectivity resolution distribution of the non-stroke controls. (**b**) The functional connectivity resolution distribution of the ischemic stroke patients.

**Figure 8 sensors-24-04234-f008:**
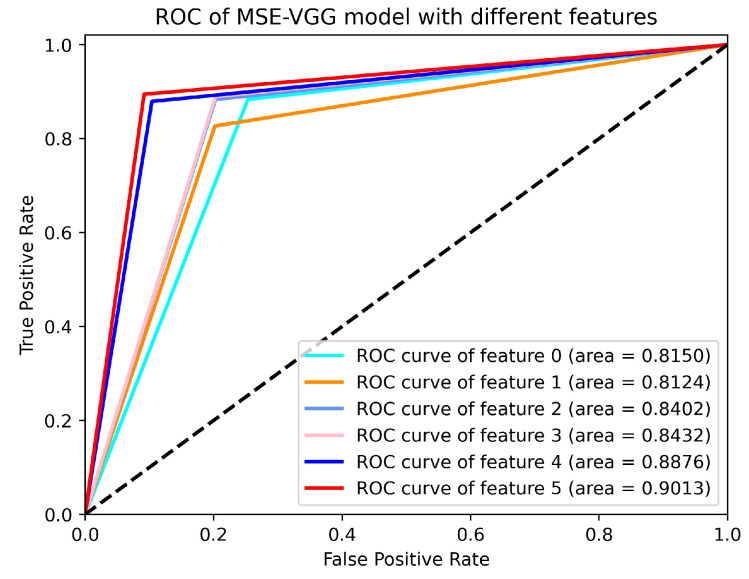
The ROC curves of the MSE-VGG classifiers with different EEG features. Features 0–5 are PCC, MI, PLV, PLI, cwPLI, and cwPLI+SaEn, respectively.

**Figure 9 sensors-24-04234-f009:**
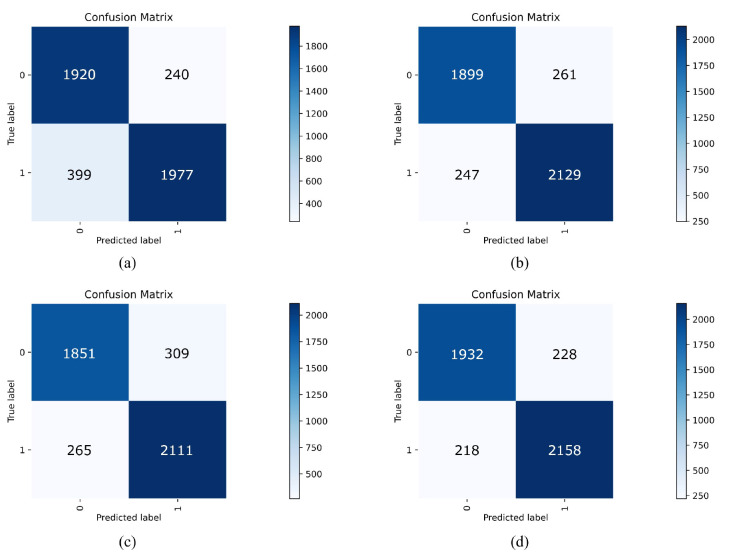
Confusion matrices of ablation experiments. (**a**) Results of cwPLI feature in VGG classifier. (**b**) Results of cwPLI feature in MSE-VGG classifier. (**c**) Results of cwPLI+SaEn feature in VGG classifier. (**d**) Results of cwPLI+SaEn feature in MSE-VGG classifier.

**Table 1 sensors-24-04234-t001:** Characteristics of ZJU4H dataset.

	Subjects	IS ${\;}^{1}$	NC ${\;}^{2}$
MRI diagnosis	39	20	19
Age—mean ± SD ${\;}^{3}$	63 ± 13	65 ± 13	60 ± 10
Sex—No. of males/total	23/39	11/20	12/19
Duration of EEG (min)	252.6	132.5	120.1

${\;}^{1}$ IS: ischemic stroke. ${\;}^{2}$ NC: non-stroke controls. ${\;}^{3}$ SD: standard deviation.

**Table 2 sensors-24-04234-t002:** The results of MSE-VGG classifiers with different EEG features.

Feature	Accuracy	Sensitivity	Specificity
PCC [23]	81.17%	76.02%	87.56%
MI [24]	81.17%	78.82%	83.52%
PLV [25]	83.82%	79.83%	88.26%
PLI [26]	84.10%	80.02%	88.61%
cwPLI	88.80%	88.49%	89.08%
cwPLI+SaEn	90.17%	89.86%	90.41%

**Table 3 sensors-24-04234-t003:** The results of the cwPLI and SaEn fusion feature with different classifiers.

Classifier	Accuracy	Sensitivity	Specificity
LR [48]	78.31%	74.67%	82.34%
AlexNet [49]	85.64%	85.41%	85.86%
RCF [50]	87.57%	84.58%	90.34%
LSTM [51]	85.98%	84.45%	87.43%
VGG [46]	87.35%	87.48%	87.23%
MSE-VGG	90.17%	89.86%	90.41%

**Table 4 sensors-24-04234-t004:** Comparison results of ablation experiments for rapid detection of ischemic stroke.

Feature Extraction	Classification	Accuracy	Sensitivity	Specificity
PLI	MSE-VGG	84.10%	80.02%	88.61%
cwPLI	VGG	85.91%	82.79%	89.17%
cwPLI	MSE-VGG	88.80%	88.49%	89.08%
cwPLI+SaEn	VGG	87.35%	87.48%	87.23%
cwPLI+SaEn	MSE-VGG	90.17%	89.86%	90.41%

**Table 5 sensors-24-04234-t005:** Feasibility analysis of comparison between traditional stroke detection methods and the proposed approach.

Method	Portability	Low Cost	Simple Operation	Time (avg)
CT	×	×	×	Within 30 min
MRI	×	×	×	More than 20 min
fMRI	×	×	×	Within 20 min
Proposed	*√*	*√*	*√*	Within 10 min + 4.62 s

## Data Availability

The ZJU4H dataset is not publicly available due to privacy or ethical restrictions.

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
