# Peer review of "MSE-VGG: A Novel Deep Learning Approach Based on EEG for Rapid Ischemic Stroke Detection"

_sensors, 2024, doi:10.3390/s24134234_

Round 1

Reviewer 1 Report

Comments and Suggestions for Authors

Authors proposed a novel method for fast ischemic stroke detection based on EEG signals analysis using the novel correlation weighted phase lag index (cwPLI) combined with sample entropy feature and using an VGG network modified with multiple squeeze and excitation modules as a classifier. Research is well-written and scientifically based. The proposed cwPLI advantages over standart PLI because of the decrease in brain functional connectivity of IS patients are proven and illustrated and is more instrumental and effective in division of stroke from control group. The final method performance demonstrated results that are superior in comparison to current analogs in accuracy, sensitivity and specificity simultaneously. The paper is good for publishing.

Reviewer 2 Report

Comments and Suggestions for Authors

Authors claim that EEG based stroke detection is new, this is not true. There are several literature that has investigated this. Please improve the introduction and those details.

Artifact removal using up to 35 Hz filtering is not justified.

Line 153 - 154 authors say 'Manual removal of EMG
artifacts and EOG artifacts increase the quality of EEG signals". Please elaborate and appropriate references.

Please explain in detail how the proposed method is the improved version of VGC.

Add Pseudo code for the proposed method.

Add ROC analysis for the results

It is not clear how the proposed method is better than CT, MRI etc. More details are needed.

Please add a detailed discussion section.

Conclusion is very weak and must be improved.

Comments on the Quality of English Language

Please get the paper proofread by a native speaker.

Reviewer 3 Report

Comments and Suggestions for Authors

This paper presents a new method for IS detection using EEG. The proposed method is discussed completely and comprehensively, compared with previous methods, the contribution of each feature used is measured, and the possibility of practical implementation is tested. This paper is very worthy of publication in Sensor.

To better explain the system being discussed, the author may need to highlight a biological explanation, such as how IS causes phase lag and the number of EEG channels used to calculate cwPLI.

Round 2

Reviewer 2 Report

Comments and Suggestions for Authors

The paper is acceptable but slight plagiarism still exists. Please fix it.

Comments on the Quality of English Language

Please get the manuscript proofread by a native speaker.